# DYNAMIC EVALUATION OF NEURAL SEQUENCE MODELS

## ABSTRACT

We present methodology for using dynamic evaluation to improve neural sequence models. Models are adapted to recent history via a gradient descent based mechanism, causing them to assign higher probabilities to re-occurring sequential patterns. Dynamic evaluation outperforms existing adaptation approaches in our comparisons. Dynamic evaluation improves the state-of-the-art word-level perplexities on the Penn Treebank and WikiText-2 datasets to 51.1 and 44.3 respectively, and the state-of-the-art character-level cross-entropies on the text8 and Hutter Prize datasets to 1.19 bits/char and 1.08 bits/char respectively.

## 1 INTRODUCTION

Sequence generation and prediction tasks span many modes of data, ranging from audio and language modelling, to more general timeseries prediction tasks. Applications of such models include speech recognition, machine translation, dialogue generation, speech synthesis, forecasting, and music generation, among others. Neural networks can be applied to these tasks by predicting sequence elements one-by-one, conditioning on the history of sequence elements, forming an autoregressive model. Convolutional neural networks (CNNs) and recurrent neural networks (RNNs), including long-short term memory (LSTM) networks (Hochreiter & Schmidhuber, 1997) in particular, have achieved many successes at these tasks. However, in their basic form, these models have a limited ability to adapt to recently observed parts of a sequence.

Many sequences contain repetition; a pattern that occurs once is more likely to occur again. For instance, a word that occurs once in a document is much more likely to occur again. A sequence of handwriting will generally stay in the same handwriting style. A sequence of speech will generally stay in the same voice. Although RNNs have a hidden state that can summarize the recent past, they are often unable to exploit new patterns that occur repeatedly in a test sequence.

This paper concerns *dynamic evaluation*, which we investigate as a candidate solution to this problem. Our approach adapts models to recent sequences using gradient descent based mechanisms. We show several ways to improve on past dynamic evaluation approaches in Section 5, and use our improved methodology to achieve state-of-the-art results in Section 7. In Section 6 we design a method to dramatically to reduce the number of adaptation parameters in dynamic evaluation, making it practical in a wider range of situations. In Section 7.4 we analyse dynamic evaluation's performance over varying time-scales and distribution shifts, and demonstrate that dynamically evaluated models can generate conditional samples that repeat many patterns from the conditioning data.

## 2 MOTIVATION

Generative models can assign probabilities to sequences by modelling each term in the factorization given by the product rule. The probability of a sequence $x_{1:T} = \{x_1, \ldots, x_T\}$ factorizes as

$$P(x_{1:T}) = P(x_1)P(x_2|x_1)P(x_3|x_2, x_1) \cdots P(x_T|x_1 \ldots x_{T-1}). \tag{1}$$

Methods that apply this factorization either use a fixed context when predicting $P(x_t|x_{1:t-1})$, for instance as in N-grams or CNNs, or use a recurrent hidden state to summarize the context, as in an RNN. However, for longer sequences, the history $x_{1:t-1}$ often contains re-occurring patterns that are difficult to capture using models with fixed parameters (static models).

In many domains, in a dataset of sequences $\{x_{1:T}^1, x_{1:T}^2, ..., x_{1:T}^M\}$, each sequence $x_{1:T}^i$ is generated from a slightly different distribution $P(x_{1:T}^i)$. At any point in time $t$, the history of a sequence $x_{1:t-1}^i$

contains useful information about the generating distribution for that specific sequence $P(x_{1:T}^i)$. Therefore adapting the model parameters learned during training $\theta_g$ is justified. We aim to infer a set of model parameters $\theta_l$ from $x_{1:t-1}^i$ that will better approximate $P(x_t^i|x_{1:t-1}^i)$ within sequence $i$.

Many sequence modelling tasks are characterised by sequences generated from slightly different distributions as in the scenario described above. The generating distribution may also change continuously across a single sequence; for instance, a text excerpt may change topic. Furthermore, many machine learning benchmarks do not distinguish between sequence boundaries, and concatenate all sequences into one continuous sequence. Thus, many sequence modelling tasks could be seen as having a local distribution $P_l(x)$ as well as a global distribution $P_g(x) := \int P(l)P_l(x)\,dl$. During training time, the goal is to find the best fixed model possible for $P_g(x)$. However, during evaluation time, a model that can infer the current $P_l(x)$ from the recent history has an advantage.

## 3 DYNAMIC EVALUATION

Dynamic evaluation methods continuously adapt the model parameters $\theta_g$, learned at training time, to parts of a sequence during evaluation. The goal is to learn adapted parameters $\theta_l$ that provide a better model of the local sequence distribution, $P_l(x)$. When dynamic evaluation is applied in the present work, a long test sequence $x_{1:T}$ is divided up into shorter sequences of length $n$. We define $s_{1:M}$ to be a sequence of shorter sequence segments $s_i$

$$s_{1:M} = \{s_1 = x_{1:n},\ s_2 = x_{n+1:2n},\ s_3 = x_{2n+1:3n},\ ...,\ s_M\}. \tag{2}$$

The initial adapted parameters $\theta_l^0$ are set to $\theta_g$, and used to compute the probability of the first segment, $P(s_1|\theta_l^0)$. This probability gives a cross entropy loss $\mathcal{L}(s_1)$, with gradient $\nabla\mathcal{L}(s_1)$, which is computed using truncated back-propagation through time (Werbos, 1990). The gradient $\nabla\mathcal{L}(s_1)$ is used to update the model, resulting in adapted parameters $\theta_l^1$, before evaluating $P(s_2|\theta_l^1)$. The same procedure is then repeated for $s_2$, and for each $s_i$ in the sequence as shown in Figure 1. Gradients for each loss $\mathcal{L}(s_i)$ are only backpropagated to the beginning of $s_i$, so computation is linear in the sequence length. Each update applies one maximum likelihood training step to approximate the current local distribution $P_l(x)$. The computational cost of dynamic evaluation is one forward pass and one gradient computation through the data, with some slight overhead to apply the update rule for every sequence segment.

As in all autoregressive models, dynamic evaluation only conditions on sequence elements that it has already predicted, and so evaluates a valid log-probability for each sequence. Dynamic evaluation can also be used while generating sequences. In this case, the model generates each sequence segment $s_i$ using fixed weights, and performs a gradient descent based update step on $\mathcal{L}(s_i)$. Applying dynamic evaluation for sequence generation could result in generated sequences with more consistent regularities, meaning that patterns that occur in the generated sequence are more likely to occur again.

## 4 BACKGROUND

### 4.1 RELATED APPROACHES

Adaptive language modelling was first considered for n-grams, adapting to recent history via caching (Jelinek et al., 1991; Kuhn, 1988), and other methods Bellegarda (2004). More recently, the neural cache approach (Grave et al., 2017) and the closely related pointer sentinel-LSTM (Merity et al., 2017b) have been used to for adaptive neural language modelling. Neural caching has recently been used to improve the state-of-the-art at word-level language modelling (Merity et al., 2017a).

The neural cache model learns a type of non-parametric output layer on the fly at test time, which allows the network to adapt to recent observations. Each past hidden state $h_i$ is paired with the next input $x_{i+1}$, and is stored as a tuple $(h_i, x_{i+1})$. When a new hidden state $h_t$ is observed, the output probabilities are adjusted to give a higher weight to output words that coincided with past hidden states with a large inner product $(h_t^T h_i)$.

$$P_{cache}(x_{t+1}|x_{1:t}, h_{1:t}) \propto \sum_{i=1}^{t-1} e^{(x_{i+1})} \exp(\omega h_t^T h_i), \tag{3}$$

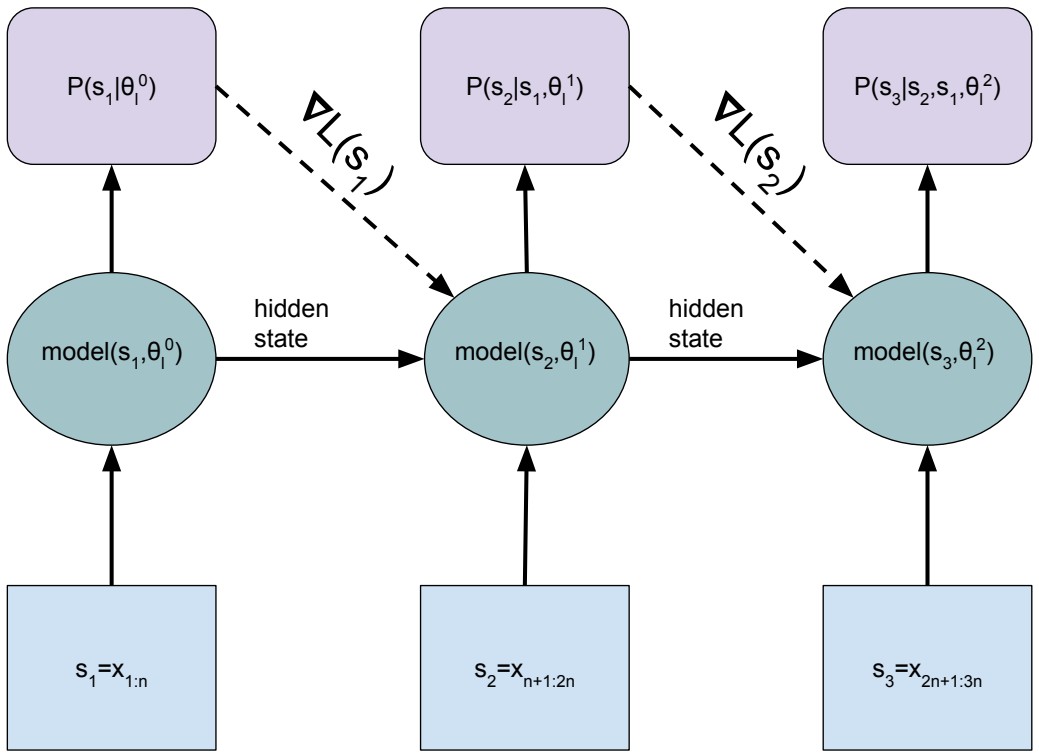

Figure 1: Illustration of dynamic evaluation. The model evaluates the probability of sequence segments $s_i$. The gradient $\nabla \mathcal{L}(s_i)$ with respect to the log probability of $s_i$ is used to update the model parameters $\theta_l^{i-1}$ to $\theta_l^i$ before the model progresses to the next sequence segment. Dashed edges are what distinguish dynamic evaluation from static (normal) evaluation.

where $e^{(x_{i+1})}$ is a one hot encoding of $x_{i+1}$, and $\omega$ is a scaling parameter. The cache probabilities are interpolated with the base network probabilities to adapt the base network at test time.

The neural cache closely relates to dynamic evaluation, as both methods can be added on top of a base model for adaptation at test time. The main difference is the mechanism used to fit to recent history: the neural cache approach uses a non-parametric, nearest neighbours-like method, whereas dynamic evaluation uses a gradient descent based method to change model parameters dynamically. Both methods rely on an autoregressive factorisation, as they depend on observing sequence elements after they are predicted in order to perform adaptation. Dynamic evaluation and neural caching methods are therefore both applicable to sequence prediction and generation tasks, but not directly to more general supervised learning tasks.

One drawback of the neural cache method is that it cannot adjust the recurrent hidden state dynamics. As a result, the neural cache's ability to capture information that occurs jointly between successive sequence elements is limited. This capability is critical for adapting to sequences where each element has very little independent meaning, e.g. character level language modelling.

Another related approach is fast weights, (Ba et al., 2016; Schmidhuber, 1992). Fast weights feature recurrent architectures with dynamically changing weight matrices as a function of recent sequence history. Thus, dynamic evaluation as applied at test time, could be considered a form of fast-weights. In traditional fast weights, the network learns to control changes to the weights during training time, allowing it to be applied to more general sequence problems including sequence labeling. In dynamic evaluation, the procedure to change the weights is automated at test time via gradient descent optimization, making it only directly applicable to autoregressive sequence modelling. As dynamic evaluation leverages gradient descent, it has the potential to generalize better to previously unseen pattern repetitions at test time.

## 4.2 Dynamic evaluation in neural networks

Dynamic evaluation of neural language models was proposed by Mikolov et al. (2010). Their approach simply used stochastic gradient descent (SGD) updates at every time step, computing the gradient with fully truncated backpropagation through time, which is equivalent to setting $n = 1$ in equation (2). Dynamic evaluation has since been applied to character and word-level language models (Graves, 2013; Krause et al., 2017; Ororbia II et al., 2017; Fortunato et al., 2017). Previous work using dynamic evaluation considered it as an aside, and did not explore it in depth.

## 5 Update Rule methodology for Dynamic evaluation

We propose several changes to Mikolov et al. (2010)'s dynamic evaluation method with SGD and fully truncated backpropagation, which we refer to as traditional dynamic evaluation. The first modification reduces the update frequency, so that gradients are backpropagated over more timesteps. This change provides more accurate gradient information, and also improves the computational efficiency of dynamic evaluation, since the update rule is applied much less often. We use sequence segments of length 5 for word-level tasks and 20 for character-level tasks.

Next, we add a global decay prior to bias the model towards the parameters $\theta_g$ learned during training. Our motivation for dynamic evaluation assumes that the local generating distribution $P_l(x)$ is constantly changing, so it is potentially desirable to weight recent sequence history higher in adaptation. Adding a global decay prior accomplishes this by causing previous adaptation updates to decay exponentially over time. The use of a decay prior for dynamic evaluation relates to the update rule used for fast weights in Ba et al. (2016), which decayed fast weights towards zero exponentially over time. For SGD with a global prior, learning rate $\eta$ and decay rate $\lambda$; we form the update rule

$$\theta_i \leftarrow \theta_{i-1} - \eta \nabla \mathcal{L}(s_i) + \lambda(\theta_g - \theta_l^{i-1}). \tag{4}$$

We then consider using an RMSprop (Tieleman & Hinton, 2012) derived update rule for the learning rule in place of SGD. RMSprop uses a moving average of recent squared gradients to scale learning rates for each weight. In dynamic evaluation, near the start of a test sequence, RMSprop has had very few gradients to average, and therefore may not be able to leverage its updates as effectively. For this reason, we collect mean squared gradients, $MS_g$, on the training data rather than on recent test data (which is what RMSprop would do). $MS_g$ is given by

$$MS_g = \frac{1}{N_b} \sum_{k=1}^{N_b} (\nabla \mathcal{L}_k)^2, \tag{5}$$

where $N_b$ is the number of training batches and $\nabla \mathcal{L}_k$ is the gradient on the $k$th training batch. The mini-batch size for this computation becomes a hyper-parameter, as larger mini-batches will result in smaller mean squared gradients. The update rule, which we call RMS with a global prior in our experiments, is then

$$\theta_l^i \leftarrow \theta_l^{i-1} - \eta \frac{\nabla \mathcal{L}(s_i)}{\sqrt{MS_g} + \epsilon} + \lambda(\theta_g - \theta_l^{i-1}), \tag{6}$$

where $\epsilon$ is a stabilization parameter. For the decay step of our update rule, we also consider scaling the decay rate for each parameter proportionally to $\sqrt{MS_g}$. Parameters with a high RMS gradient affect the dynamics of the network more, so it makes sense to decay them faster. $RMS_{\text{norm}}$ is $\sqrt{MS_g}$ divided by its mean, resulting in a normalized version of $\sqrt{MS_g}$ with a mean of 1:

$$RMS_{\text{norm}} = \frac{\sqrt{MS_g}}{\text{avg}(\sqrt{MS_g})}. \tag{7}$$

We clip the values of $RMS_{\text{norm}}$ to be no greater than $1/\lambda$ to be sure that the decay rate does not exceed 1 for any parameter. Combining the learning component and the regularization component results in the final update equation, which we refer to as RMS with an RMS global prior

$$\theta_l^i \leftarrow \theta_l^{i-1} - \eta \frac{\nabla \mathcal{L}(s_i)}{\sqrt{MS_g} + \epsilon} + \lambda(\theta_g - \theta_l^{i-1}) \odot RMS_{\text{norm}}. \tag{8}$$

## 6 SPARSE DYNAMIC EVALUATION

Mini-batching over sequences is desirable for some test-time sequence modelling applications because it allows faster processing of multiple sequences in parallel. Dynamic evaluation has a high memory cost for mini-batching because it is necessary to store a different set of parameters for each sequence in the mini-batch. Therefore, we consider a sparse dynamic evaluation variant that updates a smaller number of parameters. We introduce a new adaptation matrix $\mathcal{M}$ which is initialized to zeros. $\mathcal{M}$ multiplies hidden state vector $h_t$ of an RNN at every time-step to get a new hidden state $h'_t$, via

$$h'_t = h_t + \mathcal{M}h_t. \tag{9}$$

$h'_t$ then replaces $h_t$ and is propagated throughout the network via both recurrent and feed-forward connections. In a stacked RNN, this formulation could be applied to every layer or just one layer. Applying dynamic evaluation to $\mathcal{M}$ avoids the need to apply dynamic evaluation to the original parameters of the network, reduces the number of adaptation parameters, and makes mini-batching less memory intensive. We reduce the number of adaptation parameters further by only using $\mathcal{M}$ to transform an arbitrary subset of $H$ hidden units. This results in $\mathcal{M}$ being an $H \times H$ matrix with $d = H^2$ adaptation parameters. If $H$ is chosen to be much less than the number of hidden units, this reduces the number of adaptation parameters dramatically. In Section 7.3 we experiment with sparse dynamic evaluation for character-level language models.

## 7 EXPERIMENTS

We applied dynamic evaluation to word-level and character-level language modelling. In all tasks, we evaluate dynamic evaluation on top of a base model. After training the base model, we tune hyper-parameters for dynamic evaluation on the validation set, and evaluate both the static and dynamic versions of the model on the test set. We also consider follow up experiments that analyse the sequence lengths for which dynamic evaluation is useful.

### 7.1 SMALL SCALE WORD-LEVEL LANGUAGE MODELLING

We train base models on the Penn Treebank (PTB, Marcus et al., 1993), WikiText-2 (Merity et al., 2017b) datasets, and compare the performance of static and dynamic evaluation. These experiments compare dynamic evaluation against past approaches such as the neural cache and measure dynamic evaluation's general performance across different models and datasets.

PTB is derived from articles of the Wall Street Journal. It contains 929k training tokens and a vocab size limited to 10k words. It is one of the most commonly used benchmarks in language modelling. We consider two baseline models on PTB, a standard LSTM implementation with recurrent dropout (Zaremba et al., 2014), and the recent state-of-the-art averaged SGD (ASGD) weight-dropped LSTM (AWD-LSTM, Merity et al., 2017a).

Our standard LSTM was taken from the Chainer (Tokui et al., 2015) tutorial on language modelling[1], and used two LSTM layers with 650 units each, trained with SGD and regularized with recurrent dropout. On our standard LSTM, we experiment with traditional dynamic evaluation as applied by Mikolov et al. (2010), as well as each modification we make building up to our final update rule as described in Section 5. As our final update rule (RMS + RMS global prior) worked best, we use this for all other experiments and use "dynamic eval" by default to refer to this update rule in tables.

We applied dynamic evaluation on an AWD-LSTM (Merity et al., 2017a). The AWD-LSTM is a vanilla LSTM that combines the use of drop-connect (Wan et al., 2013) on recurrent weights for regularization, and a variant of ASGD (Polyak & Juditsky, 1992) for optimisation. Our model, which used 3 layers and tied input and output embeddings (Press & Wolf, 2017; Inan et al., 2017), was intended to be a direct replication of AWD-LSTM, using code from their implementation[2]. Results are given in Table 1.

Dynamic evaluation gives significant overall improvements to both models on this dataset. Dynamic evaluation also achieves better final results than the neural cache on both a standard LSTM and the AWD-LSTM reimplementation, and improves the state-of-the-art on PTB.

---

[1] https://github.com/chainer/chainer/tree/master/examples/ptb
[2] https://github.com/salesforce/awd-lstm-lm

| model | parameters | valid | test |
|---|---|---|---|
| RNN+LDA+kN-5+cache (Mikolov & Zweig, 2012) | | | 92.0 |
| CharCNN (Kim et al., 2016) | 19M | | 78.9 |
| LSTM (Zaremba et al., 2014) | 66M | 82.2 | 78.4 |
| Variational LSTM (Gal & Ghahramani, 2016) | 66M | | 73.4 |
| Pointer sentinel-LSTM (Merity et al., 2017b) | 21M | 72.4 | 70.9 |
| Variational LSTM + augmented loss (Inan et al., 2017) | 51M | 71.1 | 68.5 |
| Variational RHN (Zilly et al., 2017) | 23M | 67.9 | 65.4 |
| NAS cell (Zoph & Le, 2017) | 54M | | 62.4 |
| Variational LSTM + gradual learning (Aharoni et al., 2017) | 105M | | 61.7 |
| LSTM + BB tuning (Melis et al., 2017) | 24M | 60.9 | 58.3 |
| LSTM (Grave et al., 2017) | | 86.9 | 82.3 |
| LSTM + neural cache (Grave et al., 2017) | | 74.6 | 72.1 |
| **LSTM (ours)** | **20M** | **88.0** | **85.6** |
| **LSTM + traditional dynamic eval (sgd, bptt=1)** | **20M** | **78.6** | **76.2** |
| **LSTM + dynamic eval (sgd, bptt=5)** | **20M** | **78.0** | **75.6** |
| **LSTM + dynamic eval (sgd, bptt=5, global prior)** | **20M** | **77.4** | **74.8** |
| **LSTM + dynamic eval (RMS, bptt=5, global prior)** | **20M** | **74.3** | **72.2** |
| **LSTM + dynamic eval (RMS, bptt=5, RMS global prior)** | **20M** | **73.5** | **71.7** |
| AWD-LSTM (Merity et al., 2017a) | 24M | 60.0 | 57.3 |
| AWD-LSTM +neural cache (Merity et al., 2017a) | 24M | 53.9 | 52.8 |
| **AWD-LSTM (ours)** | **24M** | **59.8** | **57.7** |
| **AWD-LSTM + dynamic eval** | **24M** | **51.6** | **51.1** |

Table 1: Penn Treebank perplexities. bptt refers to sequence segment lengths.

| model | parameters | valid | test |
|---|---|---|---|
| Byte mLSTM (Krause et al., 2016) | 46M | 92.8 | 88.8 |
| Variational LSTM (Inan et al., 2017) | 28M | 91.5 | 87.0 |
| Pointer sentinel-LSTM (Merity et al., 2017b) | | 84.8 | 80.8 |
| LSTM + BB tuning (Melis et al., 2017) | 24M | 69.1 | 65.9 |
| LSTM (Grave et al., 2017) | | 104.2 | 99.3 |
| LSTM + neural cache (Grave et al., 2017) | | 72.1 | 68.9 |
| **LSTM (ours)** | **50M** | **109.1** | **103.4** |
| **LSTM + dynamic eval** | **50M** | **63.7** | **59.8** |
| AWD-LSTM (Merity et al., 2017a) | 33M | 68.6 | 65.8 |
| AWD-LSTM + neural cache (Merity et al., 2017a) | 33M | 53.8 | 52.0 |
| **AWD-LSTM (ours)** | **33M** | **68.9** | **66.1** |
| **AWD-LSTM + dynamic eval** | **33M** | **46.4** | **44.3** |

Table 2: WikiText-2 perplexities.

WikiText-2 is roughly twice the size of PTB, with 2 million training tokens and a vocab size of 33k. It features articles in a non-shuffled order, with dependencies across articles that adaptive methods should be able to exploit. For this dataset, we use the same baseline LSTM implementation and AWD-LSTM re-implementation as on PTB. Results are given in Table 2.

Dynamic evaluation improves the state-of-the-art perplexity on WikiText-2, and provides a significantly greater improvement than neural caching to both base models. This suggests that dynamic evaluation is effective at exploiting regularities that co-occur across non-shuffled documents.

| model | valid | test |
|---|---|---|
| LSTM (Grave et al., 2017) | | 121.8 |
| LSTM + neural cache (Grave et al., 2017) | | 99.9 |
| **AWD-LSTM** | **80.0** | **87.5** |
| **AWD-LSTM + neural cache** | **67.5** | **75.1** |
| **AWD-LSTM + dynamic eval** | **63.3** | **70.3** |

Table 3: text8 (word-level) perplexities

## 7.2 MEDIUM SCALE WORD-LEVEL LANGUAGE MODELLING

We benchmark the performance of dynamic evaluation against static evaluation and the neural cache on the larger text8 dataset. Like WikiText-2, text8 is derived from Wikipedia text. Text8 was introduced for word level language modelling by Mikolov et al. (2014), which preprocessed the data by mapping rare words to an '<unk>' token, resulting in a vocab of 44k and 17M training tokens. We use the same test set as in Mikolov et al. (2014), but also hold out the final 100k training tokens as a validation set to allow for fair hyper-parameter tuning (the original task did not have a validation set). We trained an AWD-LSTM with 52M parameters using the implementation from Merity et al. (2017a). We then compare the performance of static evaluation, dynamic evaluation, and neural caching at test time.

To ensure a fair comparison between dynamic evaluation and the neural cache, we used robust hyper-parameter tuning on the validation set for both methods. For dynamic evaluation, we used the hyper-parameter settings found on PTB, and only tuned the learning rate (to 2 significant figures). The neural cache uses 3 hyper-parameters: the cache length, a mixing parameter and a flatness parameter. Starting from a cache size of 3000, we used a series of grid searches to find optimal values for the mixing parameter and flatness parameter (to 2 significant figures). We then varied the cache size in the range of 2000-4000, and found that the affect of this was negligible, so we kept the cache size at 3000. Results are given in table 3, with the results from Grave et al. (2017) that used the same test set given for context.

Dynamic evaluation soundly outperforms static evaluation and the neural cache method, demonstrating that the benefits of dynamic evaluation do not wash away when using a stronger model with more training data.

## 7.3 CHARACTER-LEVEL LANGUAGE MODELLING

We consider dynamic evaluation on the character-level text8, and Hutter Prize (Hutter, 2006) datasets. The Hutter Prize dataset is comprised of Wikipedia text, and includes XML and characters from non-Latin languages. It is 100 million UTF-8 bytes long and contains 205 unique bytes. Similarly to other reported results, we use a 90-5-5 split for training, validation, and testing. The text8 dataset is derived the Hutter Prize dataset, but has all XML removed, and is lower cased to only have 26 characters of English text plus spaces. The character-level text8 task corresponds to the unprocessed version of the text8 data used for our medium-scale word level language modelling experiments. As with Hutter Prize, we use the standard 90-5-5 split for training, validation, and testing for text8. We used a multiplicative LSTM (mLSTM) (Krause et al., 2016)[3] as our base model for both datasets. The mLSTMs for both tasks used 2800 hidden units, an embedding layer of 400 units, weight normalization (Salimans & Kingma, 2016), variational dropout (Gal & Ghahramani, 2016), and ADAM (Kingma & Ba, 2014) for training.

We also consider sparse dynamic evaluation, as described in Section 6, on the Hutter Prize dataset. For sparse dynamic evaluation, we adapted a subset of 500 hidden units, resulting in a $500 \times 500$ adaptation matrix and 250k adaptation parameters. Our mLSTM only contained one recurrent layer, so only one adaptation matrix was used for sparse dynamic evaluation. All of our dynamic evaluation results in this section use the final update rule given in Section 5. Results for Hutter Prize are given in Table 4, and results for text8 are given in Table 5.

---

[3] https://github.com/benkrause/mLSTM

| model | parameters | test |
|---|---|---|
| Stacked LSTM (Graves, 2013) | 21M | 1.67 |
| Stacked LSTM + traditional dynamic eval (Graves, 2013) | 21M | 1.33 |
| Multiplicative integration LSTM (Wu et al., 2016) | 17M | 1.44 |
| HyperLSTM (Ha et al., 2017) | 27M | 1.34 |
| Hierarchical multiscale LSTM (Chung et al., 2017) | | 1.32 |
| Bytenet decoder (Kalchbrenner et al., 2016) | | 1.31 |
| LSTM + BB tuning (Melis et al., 2017) | 46M | 1.30 |
| Recurrent highway networks (Zilly et al., 2017) | 46M | 1.27 |
| Fast-slow LSTM (Mujika et al., 2017) | 47M | 1.25 |
| mLSTM (Krause et al., 2016) | 46M | 1.24 |
| **mLSTM + sparse dynamic eval ($d = 250k$)** | **46M** | **1.13** |
| **mLSTM + dynamic eval** | **46M** | **1.08** |

Table 4: Hutter Prize test set error in bits/char.

| model | parameters | test |
|---|---|---|
| Multiplicative RNN (Mikolov et al., 2012) | 5M | 1.54 |
| Multiplicative integration LSTM (Wu et al., 2016) | 4M | 1.44 |
| LSTM (Cooijmans et al., 2017) | | 1.43 |
| Batch normalised LSTM (Cooijmans et al., 2017) | | 1.36 |
| Hierarchical multiscale LSTM (Chung et al., 2017) | | 1.29 |
| Recurrent highway networks (Zilly et al., 2017) | 45M | 1.27 |
| mLSTM (Krause et al., 2016) | 45M | 1.27 |
| **mLSTM + dynamic eval** | **45M** | **1.19** |

Table 5: text8 (char-level) test set error in bits/char.

Dynamic evaluation achieves large improvements to our base models and state-of-the-art results on both datasets. Sparse dynamic evaluation also achieves significant improvements on Hutter Prize using only 0.5% of the adaptation parameters of regular dynamic evaluation.

## 7.4 TIME-SCALES OF DYNAMIC EVALUATION

We measure time-scales at which dynamic evaluation gains an advantage over static evaluation. Starting from the model trained on Hutter Prize, we plot the performance of static and dynamic evaluation against the number of characters processed on sequences from the Hutter Prize test set, and sequences in Spanish from the European Parliament dataset (Koehn, 2005).

The Hutter Prize data experiments show the timescales at which dynamic evaluation gained the advantage observed in Table 4. We divided the Hutter Prize test set into 500 sequences of length 10000, and applied static and dynamic evaluation to these sequences using the same model and methodology used to obtain results in Table 4. Losses were averaged across these 500 sequences to obtain average losses at each time step. Plots of the average cross-entropy errors against the number of Hutter characters sequenced are given in Figure 2a.

The Spanish experiments measure how dynamic evaluation handles large distribution shifts between training and test time, as Hutter Prize contains very little Spanish. We used the first 5 million characters of the Spanish European Parliament data in place of the Hutter Prize test set. The Spanish experiments used the same base model and dynamic evaluation settings as Hutter Prize. Plots of the average cross-entropy errors against the number of Spanish characters sequenced are given in Figure 2b.

On both datasets, dynamic evaluation gave a very noticeable advantage after a few hundred characters. For Spanish this advantage continued to grow as more of the sequence was processed, whereas

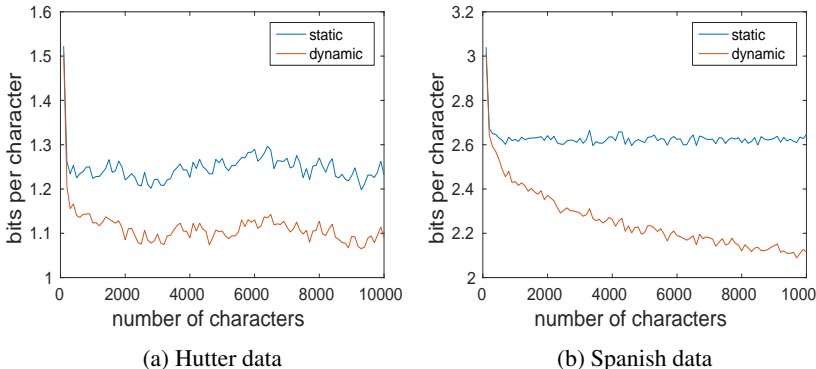

(a) Hutter data        (b) Spanish data

Figure 2: Average losses in bits/char of dynamic evaluation and static evaluation plotted against number of characters processed; on sequences from the Hutter Prize test set (left) and European Parliament dataset in Spanish (right), averaged over 500 trials for each. Losses at each data point are averaged over sequence segments of length 100, and are not cumulative. Note the different y-axis scales in the two plots.

for Hutter, this advantage was maximized after viewing around 2-3k characters. The advantage of dynamic evaluation was also much greater on Spanish sequences than Hutter sequences.

We also drew 300 character conditional samples from the static and dynamic versions of our model after viewing 10k characters of Spanish. For the dynamic model, we continued to apply dynamic evaluation during sampling as well, by the process described in Section 3. The conditional samples are given in the appendix. The static samples quickly switched to English that resembled Hutter Prize data. The dynamic model generated data with some Spanish words and a number of made up words with characteristics of Spanish words for the entirety of the sample. This is an example of the kinds of features that dynamic evaluation was able to learn to model on the fly.

## 8 CONCLUSION

This work explores and develops methodology for applying dynamic evaluation to sequence modelling tasks. Experiments show that the proposed dynamic evaluation methodology gives large test time improvements across character and word level language modelling. Our improvements to language modelling have applications to speech recognition and machine translation over longer contexts, including broadcast speech recognition and paragraph level machine translation. Overall, dynamic evaluation is shown to be an effective method for exploiting pattern re-occurrence in sequences.

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

# A APPENDIX

## A.1 DYNAMIC SAMPLES CONDITIONED ON SPANISH

300 character samples generated from the dynamic version of the model trained on Hutter Prize, conditioned on 10k of Spanish characters. The final sentence fragment of the 10k conditioning characters is given to the reader, with the generated text given in **bold**:

*Tiene importancia este compromiso en la medida en que la Comisión es un organismo que tiene el mon****tembre tas procedíns la conscriptione se ha Tesalo del Pómienda que et hanemos que Pe la Siemina.***
***De la Pedrera Orden es Señora Presidente civil, Orden de siemin presente relevante frónmida que esculdad pludiore e formidad President de la Presidenta Antidorne Adamirmidad i ciemano de el 200'. Fo***

## A.2 STATIC SAMPLES CONDITIONED ON SPANISH

300 character samples generated from the static version of the model trained on Hutter Prize, conditioned on 10k of Spanish characters. The final sentence fragment of the 10k conditioning characters is given to the reader, with the generated text given in **bold**:

*Tiene importancia este compromiso en la medida en que la Comisión es un organismo que tiene el mon****de,***
***
There is a secret act in the world except Cape Town, seen in now flat comalo and ball market and has seen the closure of the eagle as imprints in a dallas within the country." Is a topic for an increasingly small contract saying Allan Roth acquired the government in [[1916]].***

**===**

