# OpenReview forum: "Dynamic Evaluation of Neural Sequence Models"
_ICLR.cc/2018/Conference — Reject_

### Official Review · AnonReviewer2 · 2017-11-24
**Dynamic Evaluation or is it Fast Weights at test time?**

**Rating:** 7
**Confidence:** 4

**Review:**

This paper takes AWD-LSTM, a recent, state of the art language model that was equipped with a Neural Cache, swaps the cache out for Dynamic Evaluation and improves the perplexities.

Dynamic Evaluation was the baseline that was most obviously missing from the original Neural Cache paper (Grave, 2016) and from the AWD-LSTM paper. In this sense, this work fills in a gap.

Looking at the proposed update rule for Dynamic Evaluation though, the Global Prior seems to be an implementation of the Fast Weights idea. It would be great to explore that connection, or at least learn about how much the Global Prior helps.

The sparse update idea feels very much an afterthought and so do the experiments with Spanish.

All in all, this paper could be improved a lot but it is hard to argue with the strong results ...

Update:  I'm happy with how the authors have addressed these and other comments in revision 2 of the paper and I've bumped the rating from 6 to 7.

---

> ### Author Response · Authors · 2018-01-01
> **Response to AnonReviewer2**
>
> Thanks for your review.
>
> “Looking at the proposed update rule for Dynamic Evaluation though, the Global Prior seems to be an implementation of the Fast Weights idea. It would be great to explore that connection, or at least learn about how much the Global Prior helps.”
>
> This reviewer makes an insightful comment about the relationship between dynamic evaluation and fast weights. Dynamic evaluation in general does relate to fast weights (and could even be considered a type of fast weights, although it differs from traditional fast weights in the update mechanism), and the global prior we use is similar to the decay sometimes used in fast weights. We added a paragraph about this in the related work section, and updated the paper to mention the relationship with the decay rule of fast weights when we introduce the global prior.
>
> We did provide some experiments exploring how much the global prior helps in the submitted version of the paper in table 1; a simple L2 global prior helps slightly, and scaling decay rates by RMS gradient values helps a bit more.
>
> “The sparse update idea feels very much an afterthought and so do the experiments with Spanish.”
>
> We included the sparse update idea to address the high memory cost of dynamic evaluation when mini-batching. We included the Spanish experiments to show how dynamic evaluation handles domain adaptation, and as an illustration of the features that dynamic evaluation can learn to model on the fly.

---

### Official Review · AnonReviewer1 · 2017-11-25
**The proposed improvement to the dynamic evaluation method yields some performance gains on the considered benchmarks, but the authors are missing an evaluation on a larger data set.**

**Rating:** 7
**Confidence:** 4

**Review:**

The authors provide an improved implementation of the idea of dynamic evaluation, where the update of the parameters used in the last time step proposed in (Mikolov et al. 2010) is replaced with a back-propagation through the last few time steps, and uses  RMSprop rather than vanilla SGD. The method is applied to word level and character level language modeling where it yields some gains in perplexity. The algorithm also appears able to perform domain adaptation, in a setting where a character-level language model trained mostly on English manages to quickly adapt to a Spanish test set.

While the general idea is not novel, the implementation choices matter, and the authors provide one which appears to work well with recently proposed models. The character level experiments on the multiplicative LSTM make the most convincing point, providing a significant improvement over already good results on medium size data sets. Figure 2 also makes a strong case for the method's suitability for applications where domain adaptation is important.

The paper's weakest part is the word level language modeling section. Given the small size of the data sets considered, the results provided are of limited use, especially since the development set is used to fit the RMSprop hyper-parameters. How sensitive are the final results to this choice? Comparing dynamic evaluation to neural cache models is a good idea, given how both depend en medium-term history: (Grave et al. 2017) provide results on the larger text8 and wiki103, it would be useful to see results for dynamic evaluation at least on the former.

An indication of the actual additional evaluation time for word-level, char-level and sparse char-level dynamic evaluation would also be welcome.

Pros:
- Good new implementation of an existing idea
- Significant perplexity gains on character level language modeling
- Good at domain adaptation

Cons:
- Memory requirements of the method
- Word-level language modeling experiments need to be run on larger data sets

(Edit: the authors did respond satisfactorily to the original concern about the size of the word-level data set)

---

> ### Author Response · Authors · 2018-01-01
> **Response to AnonReviewer1**
>
> Thanks for your review.
>
> This reviewer requested that we add a word-level experiment on a larger dataset, and we are pleased to say that we were able to include experiments on word-level text8 in section 7.2 of the updated paper. In summary, we achieved test perplexities of static eval: 87.5, neural cache: 75.1, dynamic eval: 70.3 .These results show that dynamic evaluation still provides a large improvement to word-level language modelling on a larger dataset.
>
> Another point about the word-level experiments is that WikiText-103 and WikiText-2 use the same test set. Our result of 44.3 on the WikiText test set (using WikiText-2 for training) outperforms the static model on WikiText-103 from the neural cache paper (Grave et al. 2017) , which achieves a perplexity of 48.7.  Our result also approaches the performance of LSTM+neural cache on WikiText-103 from (Grave et al. 2017), which achieved a perplexity of 40.8. So despite using 50 times less data, our results on WikiText-2 are competitive with previous approaches trained on WikiText-103.
>
> As for the memory requirements of the method, we did present a result for character-level language modelling with our sparse dynamic evaluation that used 0.5% of the number of adaptation parameters of regular dynamic evaluation.

---

### Official Review · AnonReviewer3 · 2017-11-27
**Review of "dynamic evaluation of neural sequence models"**

**Rating:** 3
**Confidence:** 3

**Review:**

This paper proposes a dynamic evaluation of recurrent neural network language models by updating model parameters with certain segment lengths.

Pros.
- Simple adaptation scheme seems to work, and the paper also shows (marginal) improvement from a conventional method (neural cache RNNLM)
Cons.
- The paper is not well written due to undefined variables/indexes, confused explanations, not clear explanations of the proposed method in abstract and introduction (see the comments below)
- Although the perplexity is an important measure, it’s better to show the effectiveness of the proposed method with more practical tasks including machine translation and speech recognition.

Comments:
- Abstract: it is difficult to guess the characteristics of the proposed method only with a term “dynamic evaluation”. It’s better to explain it in more detail in the abstract.
- Abstract: It’s better to provide relative performance (comparison) of the numbers (perplexity and bits/char) from conventional methods.
- Section 2: Some variables are not explicitly introduced when they are appeared including i, n, g, and l
- Section 3: same comment with the above for M. Also n is already used in Section 2 as a number of sequences.
- Section 5. Why does the paper only provide examples for SGD and RMSprop? Can we apply it to other optimization methods including Adam and Adadelta?
- Section 6, equation (9): is this new matrix introduced for every layer? Need some explanations.
- Section 7.1: It’s better to provide the citation of Chainer.
- Section 7.1 “AWD-LSTM”: The paper should provide the full name of AWD-LSTM when it is first appeared.

---

> ### Comment · AnonReviewer2 · 2017-12-05
> **Re: perplexity vs MT and ASR**
>
> Vanilla neural machine translation can be viewed as conditional language modelling (most often trained for perplexity) with additional evaluation noise in the form of BLEU. Until we find a better loss, language modelling is a very good option to explore new models, optimization and evaluation techniques.
>
> The paper has issues that can be fixed up but it also has great results. It is far from a clear rejection.

---

> ### Author Response · Authors · 2018-01-01
> **Response to AnonReviewer3**
>
> Thank you for your review.
>
> This reviewer makes unreasonable claims that the improvements in the paper are “marginal”, and that language modelling is not a sufficient benchmark. As pointed out by AnonReviewer2 in the comments of this review, the results in this paper are quite strong, and language modelling is very sensible for evaluating new techniques. This reviewer also claims that the paper is not well-written, but provides very little evidence to support this.  Overall, this reviewer has no valid scientific criticisms of the paper.
>
> "the paper also shows (marginal) improvement from a conventional method (neural cache RNNLM)"
>
> Our results on WikiText-2, where we demonstrate a 7.7 perplexity point improvement over the previous state-of-the-art, would be considered far more than a "marginal" improvement by almost any standard. For instance, we report much larger perplexity gains on WikiText-2 as compared with contemporary ICLR submissions such as [1,2,3], which all use similar baselines to our experiments. Our improvements to character-level language modelling were also far more than "marginal".
>
> "Some variables are not explicitly introduced when they are appeared including i, n, g, and l"
>
>  g and l are not variables, they are subscripts used to denote "global" and "local".  If the reviewer thought g and l were variables, we can understand why the reviewer may have been confused by our explanations of our method. However, we did explicitly introduce all variables that use g and l subscripts, so this really should have been clear to the reviewer.
>
> The reviewer also mentions a few minor variables that are not “explicitly introduced”, however every one of these variables is defined implicitly in sequence/set notation.
>
> " n is already used in Section 2 as a number of sequences."
>
> To avoid confusion, we replaced n in section 2 with M, since we also use M in section 3 as the number of sequences in a slightly different context.
>
> “Abstract: it is difficult to guess the characteristics of the proposed method only with a term “dynamic evaluation”. It’s better to explain it in more detail in the abstract.”
>
>  Our use of the term dynamic evaluation was described in the second sentence of the abstract. We elected not to describe the specific engineering details of our dynamic evaluation method because this is beyond the scope of the abstract.
>
> "It’s better to provide relative performance (comparison) of the numbers (perplexity and bits/char) from conventional methods."
>
> We elected not to provide relative performance comparisons, because the "deltas" to perplexity and bits/character are almost meaningless without knowledge of how strong the baseline numbers are. Providing the static evaluation numbers alongside the dynamic evaluation numbers would also be unreasonable as it is too many results for an abstract. Providing the overall results with dynamic evaluation is the most concise way to demonstrate the effectiveness of our method, especially since all of these results improve the state-of-the-art.
>
> "Why does the paper only provide examples for SGD and RMSprop? Can we apply it to other optimization methods including Adam and Adadelta?"
>
> The goal of these experiments is to demonstrate the utility of the proposed modifications of dynamic evaluation as compared to past approaches (which used SGD). There are an infinite number of dynamic evaluation approaches, we don't claim anywhere that ours is the best possible-- just that all of our suggested modifications improve on past approaches.
>
> As for the two approaches suggested by the reviewer, ADAM or ADAM derived methods could be reasonable to use for dynamic evaluation, but were found not to work as well in preliminary experiments. Adadelta likely would not be sensible for dynamic evaluation, because the learning rates of Adadelta decrease over training.  If dynamic evaluation were applied with Adadelta, the rate of adaptation to recent history would decrease later in the test set, which would likely hurt performance.
>
> "equation (9): is this new matrix introduced for every layer? Need some explanations."
>
> The mLSTM we applied this to only used 1 recurrent layer, so this distinction was arbitrary in the context of our experiments. In a multilayer-RNN, the new matrix could be introduced for every layer or just 1 layer. We've added a sentence in the paper clarifying this point.
>
> "It’s better to provide the citation of Chainer. "
>
> we added this to the current version.
>
> “AWD-LSTM: The paper should provide the full name of AWD-LSTM when it is first appeared.”
>
> We now provide the full name of AWD-LSTM at first appearance.
>
> [1] Memory-based Parameter Adaptation. ICLR 2018 submission. https://openreview.net/pdf?id=SkFqf0lAZ
>
> [2] Breaking the Softmax Bottleneck: A High-Rank RNN Language Model. ICLR 2018 submission. https://openreview.net/pdf?id=HkwZSG-CZ
>
> [3] Fraternal Dropout. ICLR 2018 submission . https://openreview.net/pdf?id=SJyVzQ-C-

---

### Decision · Program_Chairs · 2018-01-29
**ICLR 2018 Conference Acceptance Decision**

**Decision:**

Reject

**Comment:**

The pros and cons of the paper are summarized below:

Pros:
* The proposed tweaks to the dynamic evaluation of Mikolov et al. 2010 are somewhat effective, and when added on top of already-strong baseline models improve them substantially

Cons:
* Novelty is limited. This is essentially a slightly better training scheme than the method proposed by Mikolov et al. 2010.
* The fair comparison against Mikolov et al. 2010 is only shown in Table 1, where a perplexity of 78.6 turns to a perplexity of 73.5. This is a decent gain, but the great majority of this is achieved by switching the optimizer from SGD to an adaptive method, which as of 2018 is a somewhat limited contribution. The remainder of the tables in the paper do not compare with the method of Mikolov et al.
* The paper title, abstract, and introduction do not mention previous work, and may give the false impression that this is the first paper to propose dynamic evaluation for neural sequence models, significantly overclaiming the paper's contribution and potentially misleading readers.

As a result, while I think that dynamic evaluation itself is useful, given the limited novelty of the proposed method and the lack of comparison to the real baseline (the simpler strategy of Mikolov et al.) in the majority of the experiments, I think this papers till falls short of the quality bar of ICLR.

Also, independent of this decision, a final note about perplexity as an evaluation measure to elaborate on the comments of reviewer 1. In general, perplexity is an evaluation measure that is useful for comparing language models of the same model class, but tends to not correlate well with model performance (e.g. ASR accuracy) across very different types of models. For example, see "Evaluation Metrics for Language Models" by Chen et al. 1998. The method of dynamic evaluation is similar to the cache-based language models that existed in 1998 in that it reinforces the model to choose similar vocabulary to that it's seen before. As you can see from this paper that the quality of perplexity of an evaluation measure falls when cache-based models are thrown into the mix, and one reason for this is that cache models, while helping perplexity greatly, tend to reinforce previous errors when errors do occur.